# PDC: Pearl Detection with a Counter Based on Deep Learning

**DOI:** 10.3390/s22187026

**Published:** 2022-09-16

**Authors:** Mingxin Hou, Xuehu Dong, Jun Li, Guoyan Yu, Ruoling Deng, Xinxiang Pan

**Affiliations:** 1College of Mechanical Engineering, Guangdong Ocean University, Zhanjiang 524088, China; 2Agricultural Machinery Appraisal and Extension Station in Hainan, Haikou 570206, China; 3Guangdong Marine Equipment and Manufacturing Engineering Technology Research Center, Zhanjiang 524088, China; 4South China of Marine Science and Engineering Guangdong Laboratory, Zhanjiang 524088, China

**Keywords:** object detection, pearl counting, Faster R-CNN, ResNet, noncontact

## Abstract

Pearl detection with a counter (PDC) in a noncontact and high-precision manner is a challenging task in the area of commercial production. Additionally, sea pearls are considered to be quite valuable, so the traditional manual counting methods are not satisfactory, as touching may cause damage to the pearls. In this paper, we conduct a comprehensive study on nine object-detection models, and the key metrics of these models are evaluated. The results indicate that using Faster R-CNN with ResNet152, which was pretrained on the pearl dataset, mAP@0.5IoU = 100% and mAP@0.75IoU = 98.83% are achieved for pearl recognition, requiring only 15.8 ms inference time with a counter after the first loading of the model. Finally, the superiority of the proposed algorithm of Faster R-CNN ResNet152 with a counter is verified through a comparison with eight other sophisticated object detectors with a counter. The experimental results on the self-made pearl image dataset show that the total loss decreased to 0.00044. Meanwhile, the classification loss and the localization loss of the model gradually decreased to less than 0.00019 and 0.00031, respectively. The robust performance of the proposed method across the pearl dataset indicates that Faster R-CNN ResNet152 with a counter is promising for natural light or artificial light peal detection and accurate counting.

## 1. Introduction

For centuries, people have been captivated by the natural beauty of pearls. With their lustrous sheen and sumptuous colors, pearls, as shown in Figure 1, are regarded as one of nature’s most glorious treasures. Pearls are created to protect mollusks against irritants, and they are processed into exquisite necklaces after cultivation. Pearl detection with a counter (PDC) is an important and necessary step during pearl processing. Automatic counting systems [1,2] for pearls reduce the human working time significantly, compared with traditional manual counting methods. A noncontact counting system for pearls is recommended to achieve clean and precise measurements [3]. In general, the computer vision-based noncontact counting system consists of two stages, object detection [4], and object counting [5].

Stage 1 involves object detection based on deep learning. Object detection is a computer technology related to computer vision and image processing that deals with detecting instances of semantic objects of a certain class (such as humans, buildings, or cars) in digital images and videos. Fortunately, however, the most successful approaches to object detection are currently extensions of deep learning-based models [6,7]. In 2021, Google released a new object detection model zoo for TensorFlow2, and then the prebuilt models and weights for a few specific models could be applied, which are as follows: single-shot multibox detector (SSD) with MobileNets v1 and v2, Faster-RCNN with ResNet 50,101 and 152, SSD ResNet 50, 101 and 152. According to the statistics from Web of Science, in the past five years, SSD MobileNet was adopted in a total of 107 papers [8,9], accounting for 43.7%, Faster R-CNN with ResNet in 77 papers, accounting for 31.4% [10,11], and SSD ResNet in 61 papers [12,13], accounting for 24.9%, which are three commonly used models, as shown in Figure 2.

Stage 2 involves object counting based on computer vision. The goal of object counting is to count the number of object instances in a single image or video sequence. It has many real-world applications, such as automatic bird counting [14], vehicle counting [15], and counting surgical instruments [16]. According to the Web of Science Core Collection Database, Figure 3 shows the statistics about the number of object counting papers published in the computer vision field. Compared with the period from 2013 to 2016, it is obvious that the relevant research results have increased significantly in the past five years. This is mainly due to the deep learning applications for computer vision, such as shrimp egg counting [17], people counting [18], and distribution line pole counting [19].

The Industry 4.0 paradigm will be the next step in the evolution of manufacturing, which is based on monitoring and pervasive data collection. However, the manufacturing industry has faced growing challenges in recent years. Computer vision can alleviate the biggest challenges. A subset of artificial intelligence allows computers to take in information from digital images and then make decisions based on that information. Mohamed et al. proposed a deep learning-based method, which was capable of detecting the defective paddy rice seedlings with the highest precision. It can facilitate a significant amount of work in agriculture, especially in paddy cultivation [20]. To promote the farming industry, Xinze Zheng et al. used multiple deep neural network models in the object detection and sex classification of ducks. In addition, the accuracy score of 99.29%, the F1 score of 98.60%, and 269.68 fps were achieved [21]. With the advance of Industry 4.0 and deep learning technology, Haonan Yin et al. proposed an adaptive and robust calculating algorithm to greatly save labor and material costs, which successfully facilitates inventories for stacked goods in the stereoscopic warehouse [22].

PyTorch and TensorFlow are the two most popular deep learning frameworks in 2022 by far. TensorFlow has been the go-to framework for deployment-oriented applications since its inception, and for good reason. However, PyTorch used to be extremely lackluster from a deployment perspective, although it has worked on closing this gap in recent years. From the perspective of the ease of deployment of the object detection models, we will conduct an in-depth evaluation of the nine commonly used target detection models in the TensorFlow2 model zoo, such as Faster R-CNN, SSD MobileNet, and SSD ResNet.

Balaji et al. proposed an application of a deep convolutional network (DeepSort) for the sorting and counting of haploid seeds [23]. Similarly, to solve the problems of low capability of real-time detection and counting in traffic flow statistics, Liang Mu et al. built a Yolov5 with an improved DeepSort algorithm [24]. However, despite the efforts and advances made on applying object detection models with DeepSort, it is still slow due to the base of object counting and tracking in a frame. This study is focused on the problems such as tracking. To summarize, the main contributions of this study are as follows:(i)A model is developed to automatically recognize pearls based on CNN deep learning,(ii)The accuracy of pearl detection models is evaluated using images taken from nature and artificial light images,(iii)An efficient algorithm is presented based on CNN and a corresponding computer vision setup is developed for pearl counting in densely distributed cases.

## 2. Related Work

### 2.1. Object Detection

Deep learning-based methods have demonstrated great success in many computer vision fields, including multiple-object recognition [25,26], identifying objects using cameras [27,28] and object classification [29,30]. In general, there are two different approaches for object detection, it can either make a fixed number of predictions on a grid (one stage) or leverage a proposal network to find objects and then use a second network to fine-tune these proposals and output a final prediction (two stages). The state-of-the-art one-stage object detection methods achieve good performance on datasets such as MS COCO [31], PASCAL VOC [32] and customer personal collection [33]. One-stage detectors that can predict all the bounding boxes and class probabilities in a single pass with high inference speeds are more suitable for real-time applications. Two-stage detectors have low inference speeds for the intermediate layer, which is used to propose possible object regions, but the two-stage detectors still take the lead in accuracy. The most common examples of one-stage object detectors are YOLO [34,35], SSD [36,37], SqueezeDet [38], and DetectNet [39,40]. Two-stage methods prioritize detection accuracy, and typical models include Faster R-CNN [41,42], Mask R-CNN [43,44] and Cascade R-CNN [45,46].

### 2.2. Object Count

The goal of the deep learning-based object counting task is to count the number of object instances in a single image or video sequence. It has many real-world applications. Balakrishnan presented the Faster RCNN ResNet101 algorithm, which was used to detect the number of false ceilings [47]. A deep learning recognition and counting method for the vehicle-road collaborative information was conducted by Hongbin Jiao [48]. Zishuo Huang and Qinyou Hu proposed an improved single shot multibox detector and DeepSORT algorithms and performed numerical experiments, indicating that the ship counting system is more accurate [49]. Beibei Xu, et al. adopted Mask R-CNN with feature extraction to present a low-cost approach for livestock counting in animals in real time [50].

The object count consists of two categories, which are elaborated in the following section.

With regard to statistical images method counting (SIMC), in [51], a computer-vision-based system was used to count fish by combining the information from blob detection, a mixture of Gaussians and a Kalman filter. According to the authors, the proposed method is a feasible approach for automatic fish counting, reducing costs and boosting production, as it increases labor availability. Two of the most common methods, Faster R-CNN with InceptionV2 and single-shot multibox detector (SSD) with MobileNet, have been tested, as in [52]. In fruit statistical image counting, the accuracy of Vasconez’s system reaches up to 93% (overall for all fruits) using Faster-RCNN with InceptionV2, and 90% (overall for all fruits) using SSD with MobileNet.

With regard to object tracking and counting ID (OTC), to realize automatic detection and retrain the feature extractor for multi-object tracking, a real-time vehicle tracking counter was proposed for vehicles that combines vehicle detection and vehicle tracking algorithms to realize traffic flow detection [53]. By using the ant colony algorithm and A∗ algorithm, Zihan Jiang [54] proposed an improved A∗ algorithm and combined it with the DeepSORT algorithm and YOLOv5 algorithm to carry out variable path planning. Some of the relative works are shown in Table 1.

## 3. Methods

### 3.1. Image Acquisition and Annotation

To design an efficient machine vision-based detection and count system, every aspect of the procedure from the pearl dataset to the implementation phase should be considered carefully. Such a small pearl cannot provide enough information for detection, as it has few pixels. It would be difficult to perfectly recognize every pearl through a whole image. In this study, to collect a proper dataset, several important terms were considered, including natural and artificial light source, angle of view, and the number of pearls.

In this working environment, the pearl images were captured by a 640 × 480 maximum photo resolution computer USB webcam (Figure 4a). It covered a total of 3000 images, which are divided into 2 categories, namely natural light images and artificial light images, as listed in Table 2. After the annotation process was performed, the 3000 images in the original image dataset were randomly separated into training and test image sets. To increase the model accuracy, these images were carefully labelled (Figure 4b). After the annotation process was performed, the ratio to the total image was 9:1, 90% of the pearl images were used for training, and the remaining 10% were maintained for testing, which consisted of 13,850 pearls and 1563 pearls.

### 3.2. Detector and Counting Framework

The algorithm is intended to reduce the computational burden and increase the accuracy of the base model. The detector and counter consist of two pipeline stages, as shown in Figure 5.

(i)Detector: Faster R-CNN ResNet152v1. A pearl image was first input to the convolutional backbone that can extract image features and output a feature map. ResNet152 was used as a substructure and greatly improves the effect of target detection. The region proposal network (RPN) was introduced to generate candidate regions, and the RoI pooling and classifier layer received both the optimized candidate boxes of RPN output and Conv feature map output. Finally, the classification and bounding box regression were predicted.(ii)Counter. As shown in Figure 5, the threshold was usually between 0.7 and 0.8. If the confidence value of detected people was below the threshold, the pearls were not counted. In this paper, the default threshold value of 0.8 was used for the sake of simplicity. Then, more bounding boxes of pearls were generated, and we found that pearl counting works better with more detected boxes and higher confidence values by a series of experiments.

### 3.3. Faster R-CNN ResNet152 as Detector

In this study, the faster region-based convolutional neural network (R-CNN) ResNet 152 is used for pearl detection and classification tasks. As a two-stage detector, Faster R-CNN has a higher localization and object recognition accuracy and comprises the RPN and Fast-RCNN modules. As shown in Figure 6, Faster R-CNN with ResNet consists of 152 convolution layers. Remarkably, ResNet152 is a pretrained deep CNN algorithm that performs the feature extraction task, and the 152-layer ResNet (11.3 billion FLOPs) still has lower complexity than VGG-16/19 nets (15.3/19.6 billion FLOPs) [62]. Here, ResNet152 contains the following 6 blocks: Conv1, Con2_x, Con3_x, Con4_x, Con5_x, and a fully connected layer. Because each unit contains 3 convolution layers, the total number of ResNet-152 layers is (3 + 8 + 36 + 3) × 3 = 150. If the first convolution layer and the last full connection layer are added, the total number of ResNet-152 layers is 150 + 2 = 152 layers.

A region proposal network (RPN) takes an image (of any size) as input and outputs a set of rectangular object proposals, which is an alternative to “selective search” in Fast-RCNN. Anchor boxes are predetermined boundary boxes in the RPN with a specific height and width. RPN uses the final Conv feature map as input and performs a 3 × 3 Conv sliding window operation spatially. Nine anchor boxes are generated according to center point of the sliding window. The feature map is reshaped into the softmax layer, which applies a softmax activation function to the next reshape block. The proposals generated by the RPN and the shared convolutional features are fed into the region of interest (RoI) pooling layer. The positive samples are taken up by the proposal region, with the labelled ground-truth intersection over union (IoU) value being larger than the set threshold. Finally, the extracted characteristics are transformed into a constant feature size of 7 × 7 through RoI pooling.

The scaling of RoI includes the following steps:Producing the fixed-size feature maps from non-uniform inputs,Finding a feature map obtained from a CNN after multiple convolutions and pooling layersIndicating the index and the proposal coordinates.

We have our RoI mapped onto the feature map that takes every ROI from the input and a section of input feature map, which corresponds to that ROI and converts that feature-map section into a fixed dimension map. In this study, we applied max pooling, so this process is completed on the whole RoI matrix rather than only on the topmost layer. The output fixed dimension of the ROI pooling solely depends on the layer parameters.

All bounding boxes with high confidence in the anchor box are selected for pearl detection tasks and sent to the full convolution layer through RoI pooling to obtain the category confidence and regression box of the detected image. In this study, the value of classification = 1; therefore, the positioning module uses fully connected layers to calculate the final category confidence and target position coordinates.

### 3.4. Counting Pearl

In the core of the pearl counter code, the images are analyzed by the software, looking for detection scores, boxes and classes. As shown in Figure 7, we develop all outputs as batch tensors, which are converted to NumPy arrays, and take index to remove the batch dimension. This framework allowed the input of a pearl image patch of any size within the memory of the GPU, and we slightly modified the data input of the network to fit the pearl dataset. The output detection scores, boxes and classes are fed into the object counter, and the counting range is set up between Min_Conf_Thresh and 1. It predicts a bounding box that involves the (Xmin, Ymin), (Xmax, Ymax) coordinates and the width and height and a metric of valuation of the quality confidence score.

As a rectangle has exactly four corners, we can simply count the number of corners in the detected pearl image. The approach includes the following steps:(i)Obtain a binary pearl image. Load detected pearl image, greyscale, Gaussian blur, and Otsu’s threshold;(ii)Remove small noise. We find contours and then filter them by contour area filtering with cv2.contourArea and remove the noise by filling in the contour with cv2.drawContours;(iii)Find corners. The Shi-Tomasi Corner Detector that has already been implemented, as cv2.goodFeaturesToTrack is used for corner detection;(iv)Increase count until Max_Conf_Thresh = 1. The total number of these possible rectangles is obtained, and the interpreter can return the coordinates that are the edges of the image dimensions.

Finally, the object counter can adapt to the different detection scores and boxes and provide a high counting accuracy performance. Furthermore, the label text, boxes and the total number of detections will be obtained from the model with high precision.

### 3.5. Evaluation Metrics

#### 3.5.1. Intersection over Union, IoU

Bounding box regression is an extremely important and necessary task in pearl processing detection. Intersection over union (IoU), as shown in Figure 8, illustrates the graphical view of the equation below.
(1)$$\mathrm{IoU}=\frac{\mathrm{Area}\;\mathrm{of}\;\mathrm{Overlap}}{\mathrm{Area}\;\mathrm{of}\;\mathrm{Union}}$$

In general, when IoU≥0.5, the pearl prediction is regarded as correct. The confidence score is also calculated (see Figure 9), which reflects how likely the box contains a pearl and how accurate the bounding box is.

#### 3.5.2. The Detection of Pearl Accuracy, Recall, Precision and mAP

The detection of pearl accuracy is the ratio of correct pearl predictions to the total number of input samples.
(2)$$\mathrm{accuracy}=\frac{\mathrm{correct}\;\mathrm{pearl}\;\mathrm{predictions}\;}{\mathrm{all}\;\mathrm{pearl}\;\mathrm{predictions}}$$

The detection of pearl precision is the ratio of relevant examples (true positives) among all the examples, which are defined by the following equation:(3)$$\mathrm{precision}=\frac{\mathrm{true}\;\mathrm{positives}}{\mathrm{true}\;\mathrm{positives}+\mathrm{false}\;\mathrm{positives}}$$

On the other hand, the recall is the ratio of examples, which implicates how well we find positives.
(4)$$\mathrm{recall}=\frac{\mathrm{true}\;\mathrm{positives}}{\mathrm{true}\;\mathrm{positives}+\mathrm{false}\;\mathrm{negatives}}$$

When working with pearl detection, a common approach to evaluate the effect is used to calculate the mean average precision (mAP). The average precision (AP) of the pearl class is calculated by measuring the precision for all occurrences of the object class in the pearl test dataset.

#### 3.5.3. Loss Function

$L\left(\left\{pi\right\},\left\{ti\right\}\right)$ is the Faster R-CNN loss function that optimizes the classification of each region of interest (ROI). Our loss function for a pearl image is calculated through
(5)$$L\left(\left\{pi\right\},\left\{ti\right\}\right)=\frac{1}{Ncls}\sum_{i}Lcls(pi,p_{i}^{*})+\lambda \frac{1}{Nreg}\sum_{i}p_{i}^{*}Lreg(ti,t_{i}^{*})$$
(6)$$Lcls(pi,p_{i}^{*})=\ln\left[p_{i}p_{i}^{*}+(1-pi)(1-p_{i}^{*})\right]$$
(7)$$Lreg(ti,t_{i}^{*})=smooth_{L1}(ti,t_{i}^{*})$$
where $Lcls(pi,p_{i}^{*})$ and $Lreg(ti,t_{i}^{*})$ are classification loss and regression loss, respectively. $Lcls(pi,p_{i}^{*})$ is defined by $Lcls(pi,p_{i}^{*})=\ln\left[p_{i}p_{i}^{*}+(1-pi)(1-p_{i}^{*})\right]$, and $Lreg(ti,t_{i}^{*})$ is defined by $Lreg(ti,t_{i}^{*})=smooth_{L1}(ti,t_{i}^{*})$, where
(8)$$smooth_{L1}(x)=\left\{\begin{array}{l} 0.5\times x^{2},\;\;if\left|x\right|<1\\ \left|x\right|-0.5,\;\;otherwise \end{array}\right.$$

For the regression loss, we use $Lreg(ti,t_{i}^{*})=R(ti-t_{i}^{*})$, where R is the robust loss function (*smooth_L_*_1_). λ is a hyperparameter that controls the balance between the losses, normally set to λ=1. ti and $t_{i}^{*}$ are the positions of the predicted bounding boxes and ground-truth boxes on the feature maps, obtained by Equations (9)–(12),
(9)$$tx=\frac{\left(x-xa\right)}{\omega a},\;ty=\frac{y-ya}{ha}$$
(10)$$tw=\log(\frac{\omega }{\omega a}),\;th=\log(\frac{h}{ha})$$
(11)$$t_{x}^{*}=\frac{\left(x^{*}-xa\right)}{\omega a},\;t_{y}^{*}=\frac{\left(y^{*}-ya\right)}{ha}$$
(12)$$t_{w}^{*}=\log(\frac{\omega^{*}}{\omega a}),\;t_{h}^{*}=\log(\frac{h^{*}}{ha})$$
where each grid cell predicts a bounding box that involves the coordinates x, y, ω, and h, as well as the width and height. Each variable x, xa, and $x^{*}$ for the predicted box, anchor box, and ground-truth box are determined. A measure of IoU can determine whether two bounding boxes in the same pearl image actually show the same object or a bounding box can be compared with ground-truth annotations to calculate accuracy or precision.

#### 3.5.4. Softmax

The softmax function is used in Faster R-CNN for classification among pearl possible labels. Specifically, in multinomial logistic regression and linear discriminant analysis, the input to the function is the result of K distinct linear functions, and the predicted probability for the j′ class given a sample vector x and a weighting vector w is as follows:(13)p(y(i)=jxi;θ)=eθjTx(i)∑l=1keθTx(i)
where the composition of K linear functions x→xTw1,…,x→xTwK and among the softmax function, xTw denotes the inner product of x and w. The operation is equivalent to applying a linear operator defined by w and vectors x, thus transforming the original, probably highly dimensional, input to vectors in a K-dimensional space ℝk.

## 4. Experimental Results

The network model was run on an Intel Xeon 5218R CPU workstation with an NVIDIA GeForce GTX 2080Ti (12G) GPU, using Ubuntu 20.04.4 LTS 64-bit. The pear image object detection program was developed in TensorFlow V2.8.

### 4.1. Faster R-CNN with Object Counter Performance

In this section, we collected the pearls dataset to train and test our system, as shown in Table 2. Figure 10 shows the results of applying Faster R-CNN ResNet152 with an object counter on the input images. We can classify the pearls identified by Faster R-CNN in the image and count the number of boxes for each pearl. Figure 10a–c show the results of the model for different counting performance evaluation indicators on the pearl dataset. As shown in Figure 10, the counting accuracy is maintained when the number of pearls in the image increases, which indicates that the Faster R-CNN ResNet152 model with an object counter is suitable for counting densely distributed pearls and its accuracy is basically independent of the number of pearls in the image. Therefore, this model has high detection and counting accuracy.

### 4.2. The Visualization of Training

Figure 11 shows the classification loss, localization loss and total loss of the training process. The model was trained in 30,000 steps, and when the curve of Figure 11c was examined, the total loss decreased to 0.00044 at the end of 30,000 steps. As the number of epochs increased, the classification loss (Figure 11a) and the localization loss (Figure 11b) of the model gradually decreased to less than 0.00019 and 0.00031, respectively. This shows that the model will have high accuracy in pearl detection.

## 5. Discussion

The comparison results of the seven models on the integrated test set measured by the total loss curves are shown in Figure 12. A curve at the bottom of the total loss curves indicates a better performance. It can be observed that all the networks provided acceptable performances for pearl detection, and the final experiments show that Faster R-CNN with ResNet152 had good performance in various indicators. Furthermore, we adopted the mAP and speed as quantitative metrics to evaluate the accuracy of the nine models. Although the accuracy of the nine models in evaluating mAP@0.75IoU was high, ranging from 0.9118 to 0.9883, it was found that the identification mAP@0.75IoU = 0.9883 was similar to that observed by Yu-Ting Li (2021) [63]. In Table 3, we show how the proposed system performs with different amounts of pearls in images. The performance results of the above cases are trained from 3000 images, with the total amount of pearls being 15,413. Compared with the other eight models, the proposed Faster R-CNN ResNet152 v1 with a counter achieved 98.83% in mAP@0.75IoU, which is the best result. Additionally, the average recall (AR) of Faster R-CNN ResNet152 v1 with a counter is also better than that of the other models (recall/AR@100(small) = 0.8643). It can be observed in Table 3 that the size of Faster R-CNN ResNet152 v1 with the recognition model proposed in the study was only 15 MB, which can effectively realize the detection. The high inference speed as the impact of key factors for deep learning has been widely realized in detection [64,65,66]; however, the loading time of a model is always overlooked. Nevertheless, we cannot always depend on our model for inference in a cache, and we still need a short loading time for predictions. Table 3 shows the loading time of the model measured on a Dell 7920 workstation with Ubuntu and the newest TensorFlow 2.8 and Python 3.10. The first time that we load the model, it takes a long time because the OpenCL kernel for the GPU stack will compile. However, each subsequent inference of the same model will be much faster, and the experimental results are shown in Table 3.

The results shown in Table 3 indicate that these values of different models present no clear tendency. Therefore, the bar charts of the model loading time and model size are also plotted, as shown in Figure 13 and Figure 14. It can be noted that the loading time and side of Faster R-CNN ResNet152 v1 in this study are intermediate levels among the nine detection models. However, as the two most important indicators, the proposed Faster R-CNN ResNet 152 v1 ensures the best detection recall/AR100(small) = 0.8643 (see Figure 15). Meanwhile, the mAP@0.75IoU = 0.9883 of Faster R-CNN ResNet 152 v1 shows the second-best precision after SSD MobleNetv2, which is superior to the other seven models (see Figure 16). As pearls are valuable, it is necessary to guarantee that there are not any missing in counting. Therefore, to balance the recall and mAP, the Faster R-CNN ResNet152 v1 with a counter is more suitable for pearl auto detection and counting.

To further illustrate the effect of the three classification networks of Faster R-CNN, the overall performances of ResNet50, ResNet101 and ResNet152 were comparatively analyzed. In Figure 17, the red line is the Faster R-CNN ResNet152v1 model, and it can be clearly observed that ResNet152 has a smaller fluctuation when convergence is achieved according to the total loss curves. In conclusion, compared with ResNet50 and ResNet101, Faster R-CNN ResNet152 has better detection and recognition performance for pearls in natural sunlight and artificial light sources, so as to verify that the Faster R-CNN with ResNet152 is optimal.

## 6. Conclusions and Future Work

An accurate and hygienic PDC method is necessary for pearl production management. Current solutions for pearl counting mainly rely on manual counting, which is generally not accessible to small and medium producers. In this study, a new pearl counting technique is introduced, and the model is composed of object detection and a counting algorithm. Through the comparison of nine object models, encouraging results are obtained. The accuracy of mAP@0.75IoU of Faster R-CNN ResNet152v1 is 98.83%, which achieves the expected effect. Moreover, the results show that the pearl detection has favorable robustness, which indicates that this method can be extended to the rapid detection of the pearls in natural light and artificial light environments, if appropriate training datasets are available. The proposed deep learning network achieved a recall/AR@100(medium) of 95% in pearl identification, and 100% accuracy in pearl counting. Finally, it is of great significance in the exploration of noncontact and high-throughput object counting. There are mainly two tasks that should be carried out in future work: to evaluate and compare more object detection models and to measure the object sizes and reduce these method limitations.

## Figures and Tables

**Figure 1 sensors-22-07026-f001:**
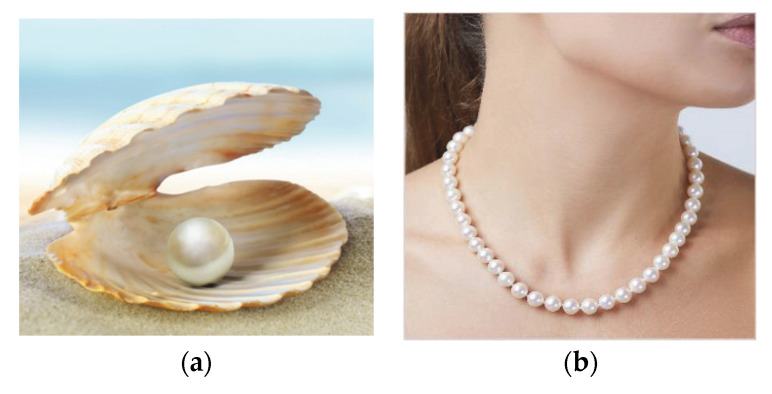
The finest quality of pearls. (**a**) Pearl in a mollusk shell, (**b**) pearl necklace.

**Figure 2 sensors-22-07026-f002:**
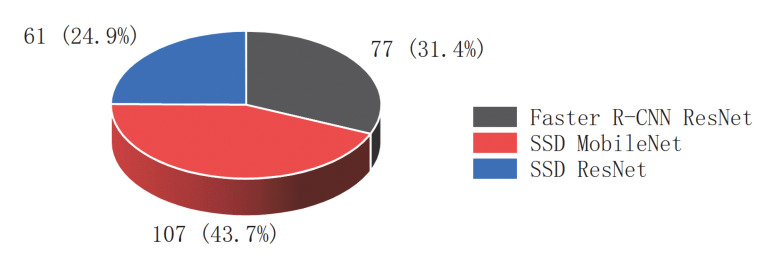
The application proportions of the three models from 2018 to 2022, including the Faster R-CNN ResNet, SSD MobileNet and SSD ResNet models. (From Web of Science Core Collection).

**Figure 3 sensors-22-07026-f003:**
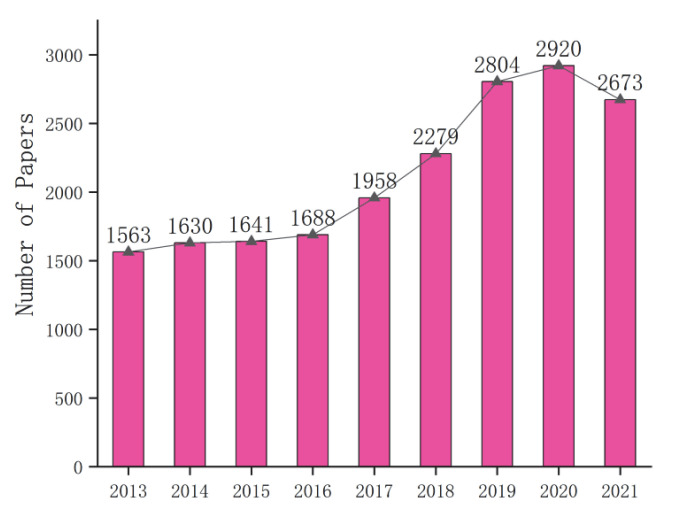
The number of object counting papers published in the field of computer vision (from Web of Science).

**Figure 4 sensors-22-07026-f004:**
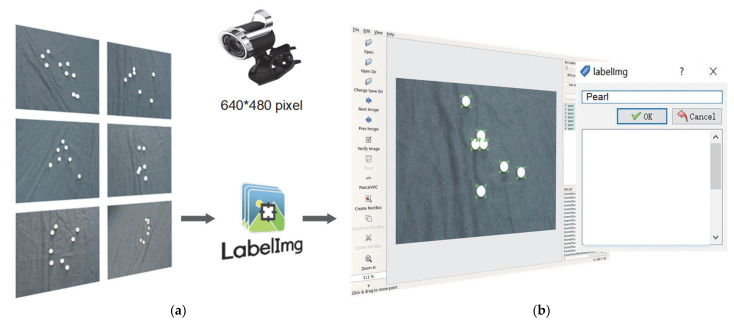
Illustration of the image capturing process and annotations. (**a**) Pearl image collections. (**b**) Annotations by LabelImg.

**Figure 5 sensors-22-07026-f005:**
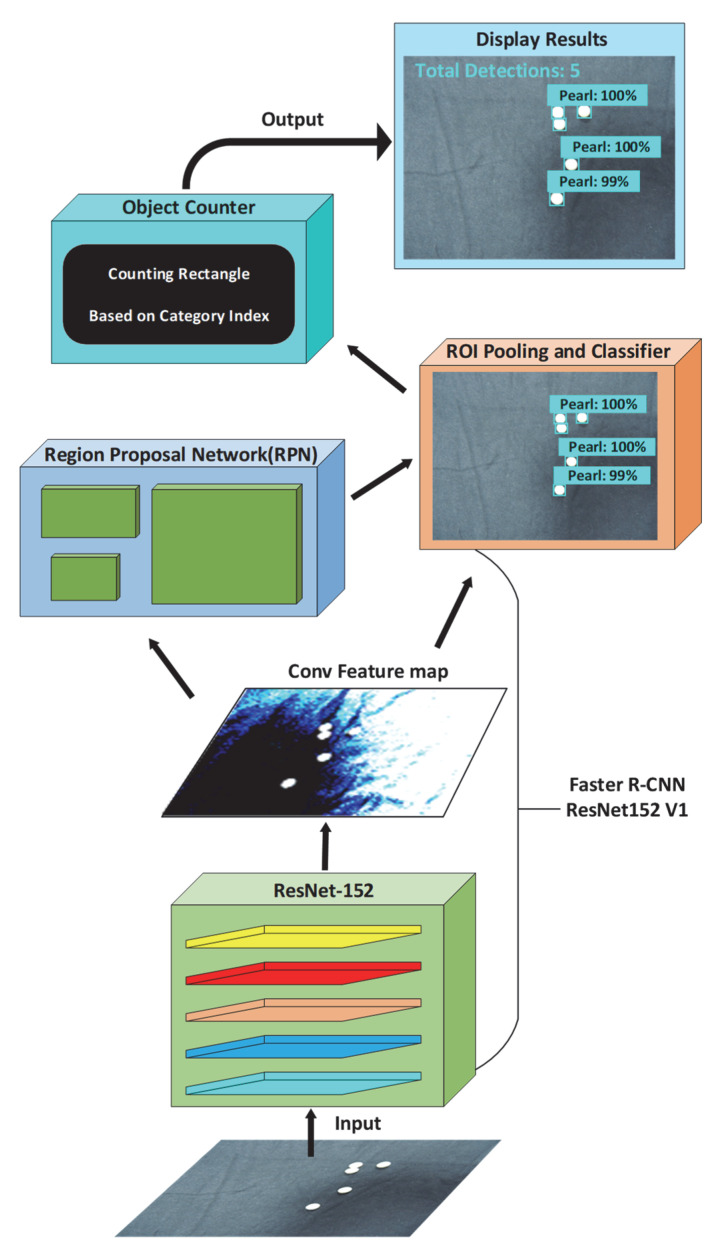
The framework of Faster R-CNN with a counter.

**Figure 6 sensors-22-07026-f006:**
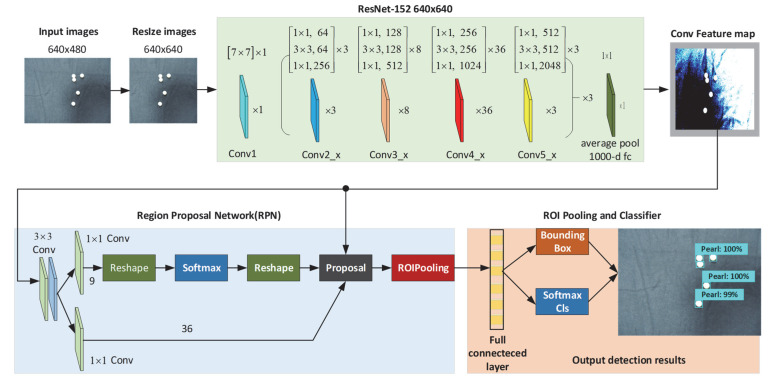
The original Faster R-CNN with RestNet152.

**Figure 7 sensors-22-07026-f007:**
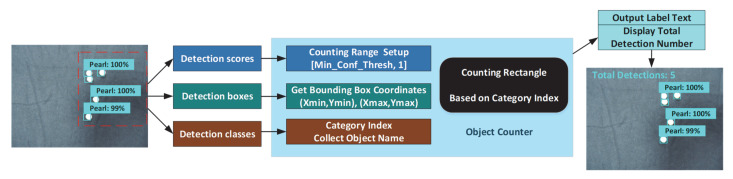
Object counter architecture of pearls.

**Figure 8 sensors-22-07026-f008:**
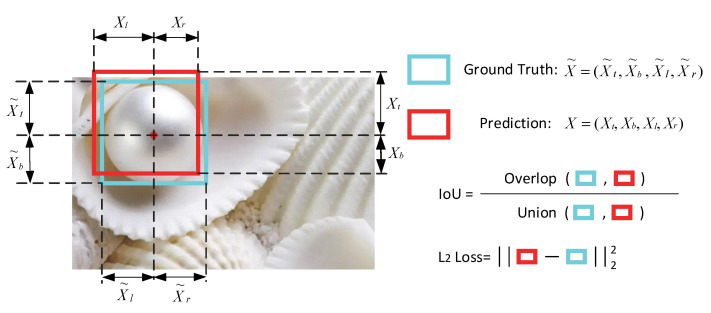
IoU.

**Figure 9 sensors-22-07026-f009:**
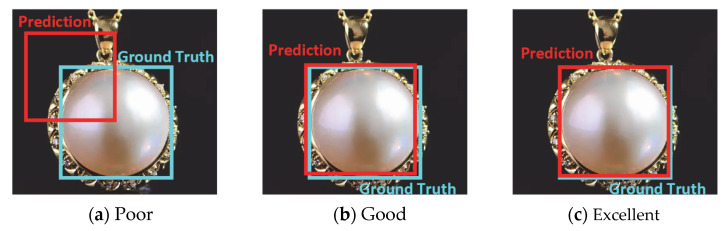
The higher the detection IoU, the better the performance. (**a**) IoU = 0.396. (**b**) IoU = 0.886. (**c**) IoU = 0.9838.

**Figure 10 sensors-22-07026-f010:**
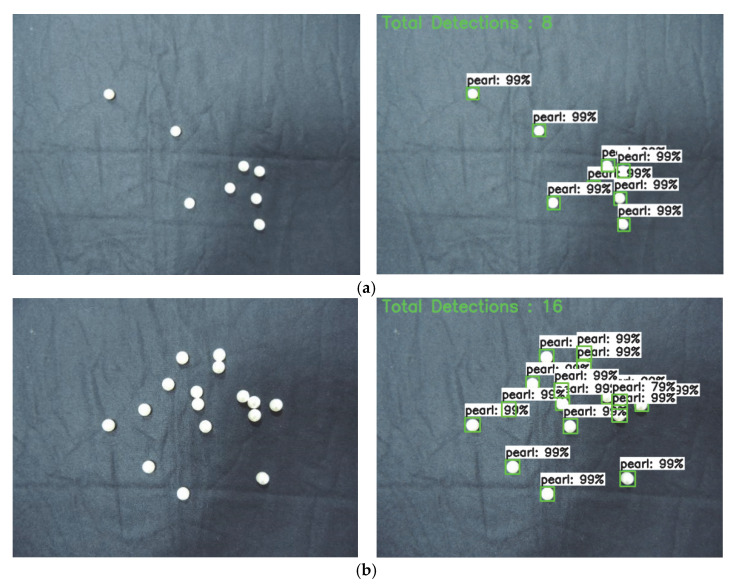
Partial prediction results of Faster R-CNN ResNet152 with an object counter. (**a**) The total number of detected pearls is 8. (**b**) The total number of detected pearls is 16. (**c**) The total number of detected pearls is 20.

**Figure 11 sensors-22-07026-f011:**
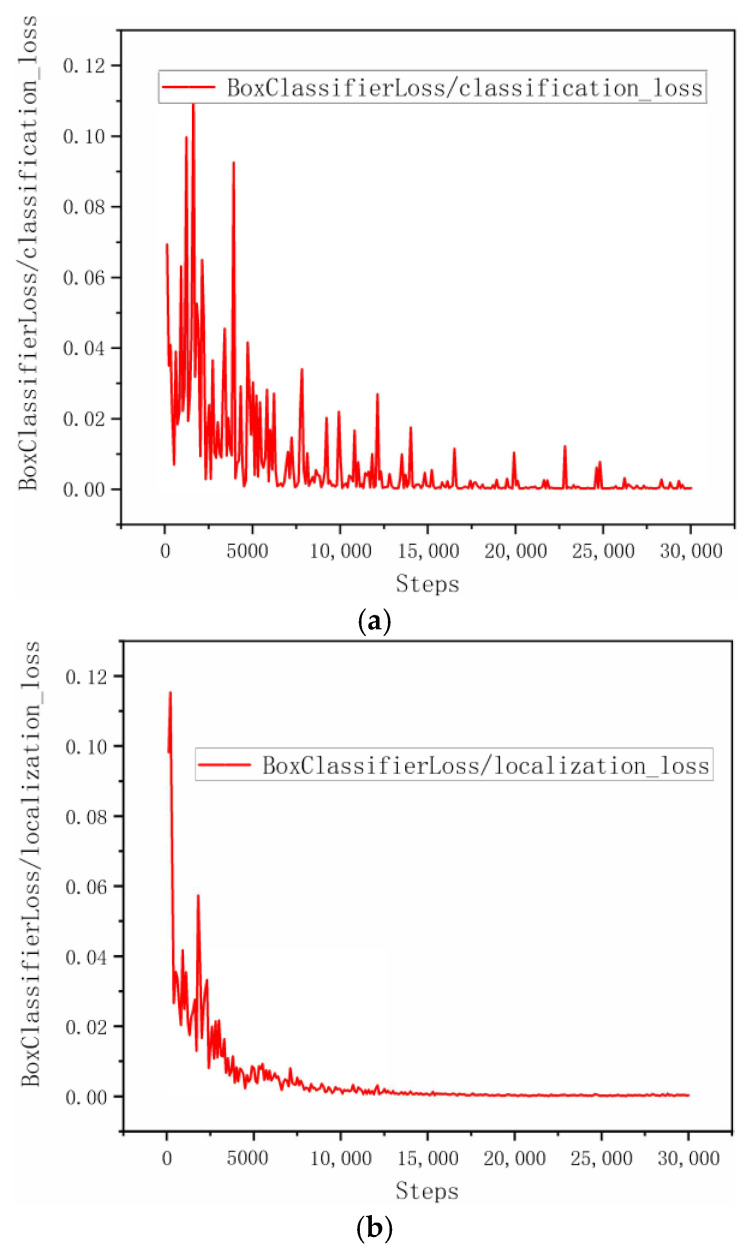
The loss curve of Faster R-CNN with ResNet152: (**a**) classification loss; (**b**) localization loss; (**c**) total loss.

**Figure 12 sensors-22-07026-f012:**
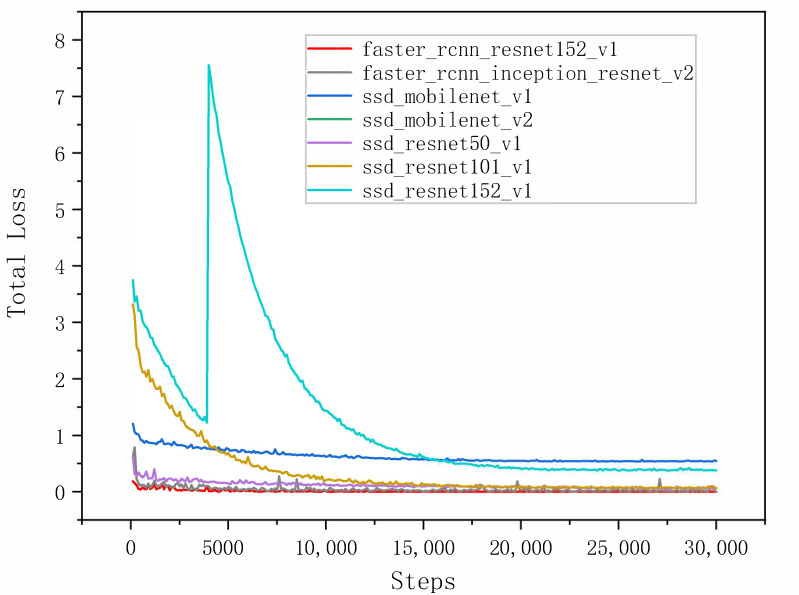
The total loss curves of all algorithms (Faster R-CNN ResNet 152, Faster R-CNN Inception ResNet, SSD MobileNet v1, SSD MobileNet v2, SSD ResNet50, SSD ResNet101 and SSD ResNet152).

**Figure 13 sensors-22-07026-f013:**
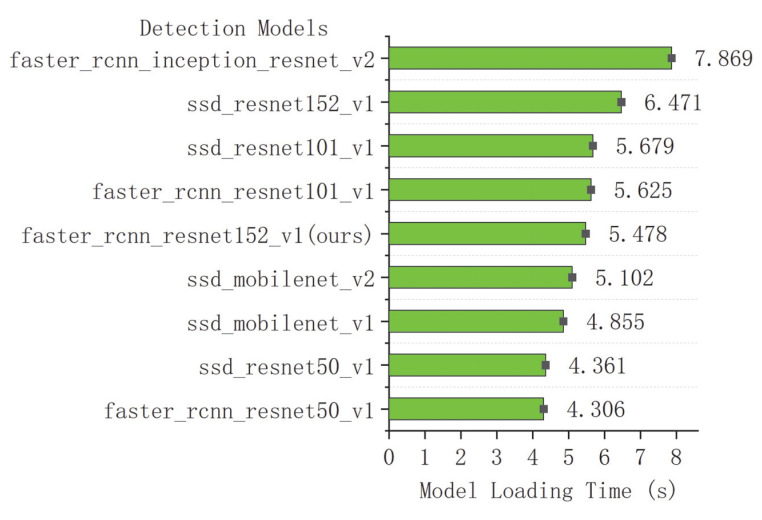
Comparison of detection models in loading time.

**Figure 14 sensors-22-07026-f014:**
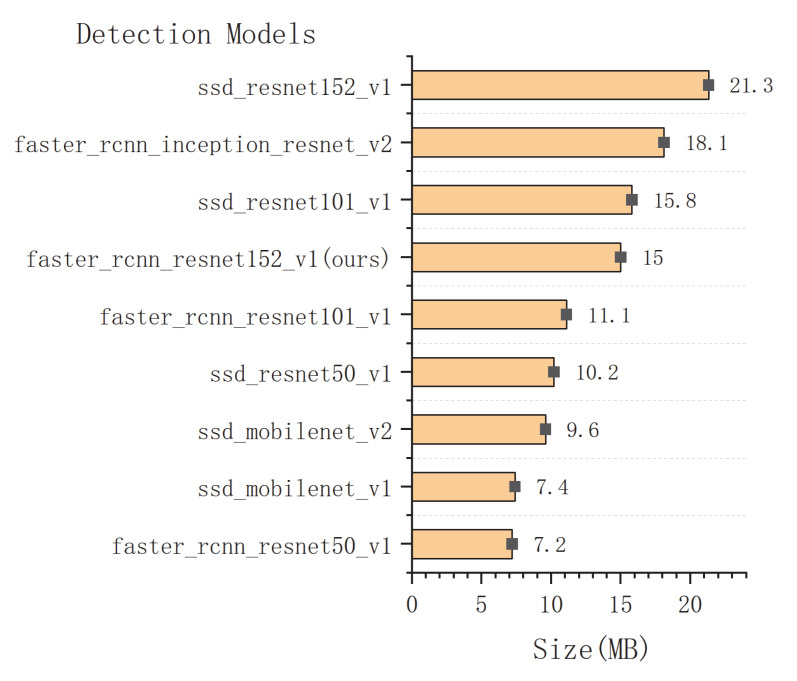
Comparison of detection models in sizes.

**Figure 15 sensors-22-07026-f015:**
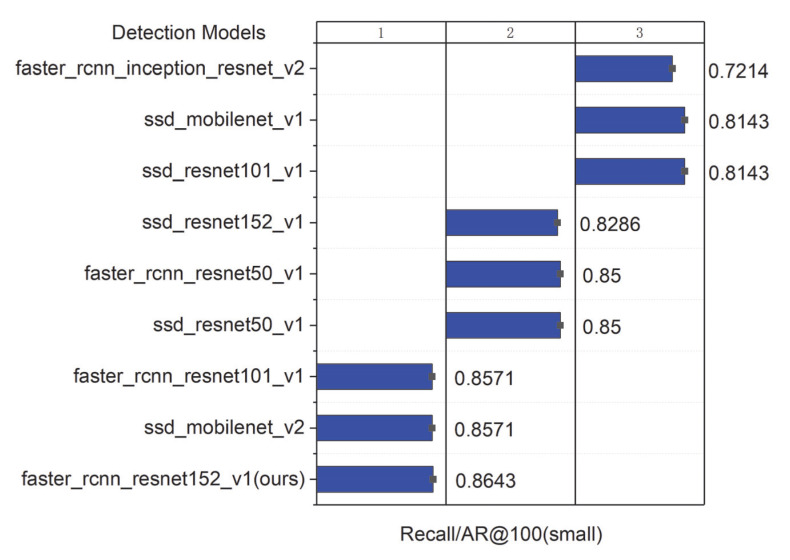
Values of Recall/AR100(small) in different detection models.

**Figure 16 sensors-22-07026-f016:**
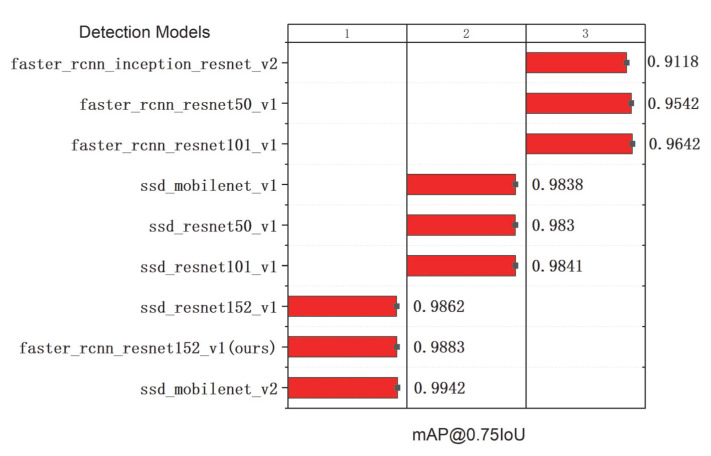
Values of mAP@0.75IoU in different detection models.

**Figure 17 sensors-22-07026-f017:**
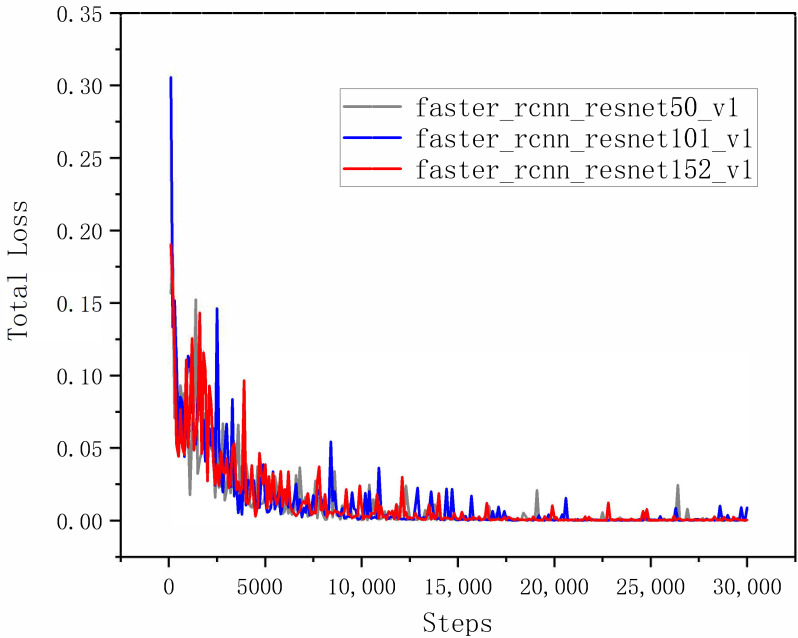
Comparison of total loss curves among Faster R-CNN with ResNet: ResNet50, ResNet101 and ResNet152.

**Table 1 sensors-22-07026-t001:** Some of the representative studies in recent years.

Num	Authors	Organization	Fundamentals	Technologies	Application	Year
1	Xiaoning Yu [55]	China Agricultural University	Deep learning	Multi-modules and attention mechanism	Aquaculture	2022
2	Wan Nurazwin R. [56]	Shah Alam Selangor	Image recognition	Machine learning classifiers	Pineapple crown	2021
3	Mark F. Hansen [57]	UWE Bristol	Machine vision	Black Soldier Fly (BSF)	insect farming	2022
4	Md Sultan Mahmud [58]	The Pennsylvania State University	Deep learning	LSTM + Mask RCNN	Cattle agriculture	2021
5	J.P. Vasconz [52]	Universidad Técnica Federico Santa María	CNN	Faster-RCNN with Inception V2	Fruit counting	2020
6	Yue Mu [59]	Nanjing Agricultural University	Deep learning	R-CNN with ResNet101	Tomato counting	2020
7	Sylvia T. Kouyoumdjieva [60]	KTH Royal Institute of Technology	No-image recognition	RSSI + CSI	People counting	2020
8	Mihir Durve [61]	Fondazione Istituto Italiano di Tecnologia	Deep learning	YOLO + DeepSORT	Track moving droplets	2021

**Table 2 sensors-22-07026-t002:** Detailed information of pearl image sets.

Image Set	Amount of Images	Amount of Pearls	Image Conditions	
Train image set	2700	13,850	Natural light images	Artificial light images
			1350	1350
Test image set	300	1563	Natural light images	Artificial light images
			150	150

**Table 3 sensors-22-07026-t003:** Metrics comparison of different models.

Model	mAP @0.5IoU	mAP @0.75IoU	Recall/AR @100(medium)	Recall/AR @100(small)	Model Loading Time	Speed Inference Time with Counter	Size
Faster R-CNN ResNet152 v1 (ours)	1	0.9883	0.95	0.8643	5.478 s	15.8 ms	15 MB
Faster R-CNN ResNet101 v1	1	0.9642	0.9325	0.8571	5.625 s	14.9 ms	11.1 MB
Faster R-CNN ResNet50 v1	1	0.9542	0.9312	0.85	4.306 s	15.8 ms	7.2 MB
Faster R-CNN Inception ResNet v2	1	0.9118	0.925	0.7214	7.869 s	16.0 ms	18.1 MB
SSD MobileNet v1	0.9978	0.9838	0.94	0.8143	4.855 s	14.5 ms	7.4 MB
SSD MobileNet v2	1	0.9942	0.9375	0.8571	5.102 s	15.1 ms	9.6 MB
SSD ResNet50 v1	1	0.9830	0.925	0.85	4.361 s	13.4 ms	10.2 MB
SSD ResNet101 v1	0.9841	0.9841	0.9	0.8143	5.679 s	15.2 ms	15.8 MB
SSD ResNet 152 v1	0.9862	0.9862	0.8875	0.8286	6.471 s	15.5 ms	21.3 MB

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
