# Peer review of "PDC: Pearl Detection with a Counter Based on Deep Learning"

_sensors, 2022, doi:10.3390/s22187026_

Round 1

Reviewer 1 Report

1 The English writing of this paper should be highly improved. 

2 Many of references cited after some public applications, e.g. vehicle and pedestrian counting [53,54], are only used to increase the reference number without being introduced.

3 This paper only establish a pearl dataset and apply popular detection models to conduct detection and counting. The novelty is not enough. 

Reviewer 2 Report

In many industrial applications, pearl-type objects can be counted using much simpler methods.

In the introduction, it would be advisable to present in more detail the conditions of the industrial applications in which the detection is done.

In addition to detecting and counting the pearls, it would also be useful to measure their sizes.

Reviewer 3 Report

- The authors have to improve the article grammatically, There are many grammatical errors in the article.

- The abstract can include the experimental results obtained.

- "PyTorch and TensorFlow are far away from the two most popular deep learning frameworks in 2022"

is it "Pytorch and tensorflow are by far...."?? Pls check

- Enhance the introduction section by providing a discussion on teh motivations, novelty of this study.

- Summarize the related works in the form of a table. SOme of the recent works such as the following can be discussed : Computer Vision and Recognition Systems Using Machine and Deep Learning Approaches: Fundamentals, Technologies and Applications

- How did the authors choose the hyper-parameters for R-CNN? is it random or by using any hyper parameter tuning approaches? If its random it is suggested that the authors do experimentation with hyper-parameter tuning approaches too.

- What is the computational cost of the proposed R-CNN approach?

- What are the modifications that are required in the architecture so that it can be extended to the pearl detection and counting in real time?

-  What are the threats to validity of the proposed approach?

- Discuss the limitations of this work in conclusion.

Round 2

Reviewer 1 Report

The authors have answered my concerns. The English should be further improved.

Reviewer 3 Report

The authors have addressed the comments well. The article can be accepted for publication.